# The effect of bio-banding on academy soccer player passing networks: Implications of relative pitch size

**Christopher Towlson**[1]*, **Grant Abt**[1], **Steve Barrett**[2], **Sean Cumming**[3], **Frances Hunter**[4], **Ally Hamilton**[5], **Alex Lowthorpe**[1], **Bruno Goncalves**[6,7,8], **Martin Corsie**[9], **Paul Swinton**[9]

**1** Department of Sport, Health and Exercise Science, University of Hull, Hull, United Kingdom, **2** Playermaker, London, United Kingdom, **3** Department for Health, University of Bath, Bath, United Kingdom, **4** Middlesbrough Football Club, Middlesbrough, United Kingdom, **5** Hull City Football Club, Hull, United Kingdom, **6** Departamento de Desporto e Saúde, Escola de Saúde e Desenvolvimento Humano, Universidade de Évora, Évora, Portugal, **7** Comprehensive Health Research Centre (CHRC), Universidade de Évora, Évora, Portugal, **8** Portugal Football School, Portuguese Football Federation, Oeiras, Portugal, **9** School of Health Sciences, Robert Gordon University, Aberdeen, United Kingdom

* c.towlson@hull.ac.uk

**Data Availability Statement:** All anonymised and raw data is accessible via the University of Hull open access repository. https://hull-research.worktribe.com/record.jx?recordid=3796271.

## Abstract

The primary aims of this study were to examine the effects of bio-banding players on passing networks created during 4v4 small-sided games (SSGs), while also examining the interaction of pitch size using passing network analysis compared to a coach-based scoring system of player performance. Using a repeated measures design, 32 players from two English Championship soccer clubs contested mixed maturity and bio-banded SSGs. Each week, a different pitch size was used: Week 1) small (36.1 $m^2$ per player); week 2) medium (72.0 $m^2$ per player); week 3) large (108.8 $m^2$ per player); and week 4) expansive (144.50 $m^2$ per player). All players contested 12 maturity (mis)matched and 12 mixed maturity SSGs. Technical-tactical outcome measures were collected automatically using a foot-mounted device containing an inertial measurement unit (IMU) and the Game Technical Scoring Chart (GTSC) was used to subjectively quantify the technical performance of players. Passing data collected from the IMUs were used to construct passing networks. Mixed effect models were used with statistical inferences made using generalized likelihood ratio tests, accompanied by Cohen's local $f^2$ to quantify the effect magnitude of each independent variable (game type, pitch size and maturation). Consistent trends were identified with mean values for all passing network and coach-based scoring metrics indicating better performance and more effective collective behaviours for early compared with late maturation players. Network metrics established differences ($f^2$ = 0.00 to 0.05) primarily for early maturation players indicating that they became more integral to passing and team dynamics when playing in a mixed-maturation team. However, coach-based scoring was unable to identify differences across bio-banding game types ($f^2$ = 0.00 to 0.02). Pitch size had the largest effect on metrics captured at the team level ($f^2$ = 0.24 to 0.27) with smaller pitch areas leading to increased technical actions. The results of this study suggest that the use of passing networks may provide additional insight into the effects of interventions such as bio-banding and that the number of early-maturing players should be considered when

**Funding:** The author(s) received no specific funding for this work.

**Competing interests:** The authors have declared that no competing interests exist.

using mixed-maturity playing formats to help to minimize late-maturing players over-relying on their early-maturing counterparts during match-play.

## Introduction

Growth and maturation confound the identification and development of talented, adolescent soccer players [1–3] because the tempo of anthropometric development is asynchronous with age [4, 5] leading to phases of accelerated growth rates onsetting at 9–11 years and ceasing at 13–15 years [4, 5]. This period of temporary accelerated growth in stature is typically referred to as peak height velocity (PHV) [6–8]. The non-linear relationship between stature development and age can create large variations in anthropometric [9] and key physical fitness characteristics within chronologically aged groups [9–13]. These maturity-related differences can lead to sub-conscious maturation selection bias resulting in over-selection of early maturing players who possess temporary enhancements in maturity-related anthropometric (i.e. taller) and physical fitness characteristics (e.g. superior sprinting capacity), despite that technical and psychological characteristics of players are considered key attributes for talent selectors [14].

To control for the confounding influence of maturation alone [15] during talent identification and development, researchers and practitioners have grouped players according to maturation status (typically referred to as 'bio-banding' [16, 17]) to create homogenous groups of players who are primarily 'matched' for maturity-related anthropometric characteristics. However, despite players and key stakeholders valuing the approach [18–20], there is limited applied soccer-based research to support its efficacy [21–25]. That said, Abbott, Williams [23] reported that matching players for maturity status (i.e., late maturing vs late maturing, early maturing vs early maturing) during match-play may control maturity-related differences in physical match-activity profiles, while altering the technical demands (e.g., shots, dribbles, tackles etc). In addition, Towlson, MacMaster [21] have stated that mis-matching players (i.e., late maturing vs early maturing) during bio-banded formats may enhance the identification of desirable psychological characteristics of pre-PHV academy soccer players [21]. However, they also suggested that the small, relative size of the single pitch used in the study may have limited the expression of other maturity-related match-play characteristics. This is important to practitioners responsible for identifying talented soccer players given that contextual match factors such as larger relative pitch size likely afford earlier-maturing players the opportunity to apply tactical superiority due to their transient anthropometric, physical fitness and decision-making characteristics [10, 11, 13, 26]. Such considerations are particularly important when considering player performance, as physical performance is position specific [27] and can significantly increase on a large pitch, with inter-team and intra-team distances becoming significantly larger, subsequently increasing within-team tactical variability (i.e., intra-team distances) [28].

The outcome measures used to quantify the physical, technical, tactical, and psychological behaviours constituting individual and collective performance may also influence assessment of the effects of bio-banding. Passing-related outcomes are frequently used to evaluate technical and tactical performance of players and traditionally have focussed on relatively simple metrics such as the number of pass attempts, percentage completion and number of passes per possession [29, 30]. However, over the last decade, network analysis has become one of the most active fields in quantitative analysis and the creation of passing networks is increasingly being applied in soccer to better capture the complex dynamics present [31]. While analysis of passing data can give rise to multiple types of networks comprising players and regions of the

pitch [31], passing networks have mostly been restricted to successful passes between players in each team [32]. A passing network is constructed such that each player is considered a node in the network and visually a graph can be identified where successful passes link players such that the graph can be directed (passer and receiver) and weighted (more passes create stronger links). Both team (macro) and individual-based (micro) metrics can be extracted from passing networks to quantify performance and cooperation at the macro level (structure of the network), with centrality metrics commonly used to quantify the importance and prominence of individual players [32, 33]. Passing networks have been used to assess performance in both elite competition and in training comprising SSGs [33, 34], with analyses found to provide coaches with meaningful information on player performance and the collective behaviours of the team [32]. This information can be used to adjust behaviours on the field in both attack and defence to enhance synergy between players and overall team synchronization [32].

Initial research appears to indicate changes in some performance characteristics such as technical load, perceptual effort and psychological-related behaviours during bio-banded matches when compared with matches based on chronological age [21, 23]. Authors have previously argued that network analyses can provide additional insights into the performance of youth soccer players beyond traditional notational analyses that focus on *what* happened, and instead provide understanding of more complex behaviours and *how* performance occurs [35]. Therefore, the primary aim of this study was to investigate the effects of bio-banding players on passing networks created during 4v4 SSGs. In addition, as pitch size can influence tactical actions [36–39], with smaller pitch areas ($<100$ m$^2$ per player) being shown to increase the number of technical actions [40] and larger pitches eliciting greater physical demands and more opportunity for players to record higher running speeds [41], the interaction of pitch size was also investigated. Finally, given that coach observations are often the first point of player evaluation (i.e., scouts etc), it is important to establish if coach observations provide a suitable assessment of players technical and tactical actions during match-play. Therefore, network analysis measures will be compared to subjective coach-based scoring system of player performance during bio-banded match-play.

## Materials and methods

### Participants

Following institutional ethics committee approval (FHS189) and informed, verbal parental/guardian consent, forty-four, male, UK academy soccer players (mean (SD) age 12.9 (0.9) years, body mass 46.4 (8.5) kg and stature 158.2 (14.9) cm) participated in the study. Players were initially over-recruited (to permit an adequate numbers of players per maturity banding to be identified, while accounting for attrition) from two English Championship clubs, including one in category 1 (n = 24) and one in category 2 (n = 20). Although no *a priori* sample size estimation was conducted, our sample size was determined, and constrained by, the number of academy players available across the two clubs. All players were free of injury and identified as fit to participate in the study. The players were exposed to 6 to 12 hours of coached practice on a weekly basis. Players' anthropometric data were collected for the purposes of bio-banding classification. As per previous methods [5, 10, 11, 21], a portable stadiometer (Seca 217, Chino, U.S.A) was used to measure player stature, with participants asked to stand with their feet together without wearing shoes, hold a deep breath with the head positioned in the Frankfurt plane while light vertical traction was applied. Body mass (seca robusta 813, Chino, U.S.A.) was measured while players were shoeless but wearing their usual training attire. Anthropometric data were combined with age and self-reported parental height adjusted for over-estimation [42], with player estimated percentage of parental adult height (PAH%) calculated

using the Khamis and Roche [43] method. The Khamis and Roche [43] method is commonly used within academy soccer programmes [44], often as a surrogate for more invasive measures of biological maturation (e.g., stage of pubic hair development [45, 46] and skeletal age [47]). Although we recognise that PHV onsets at approximately 86% of estimated adult stature attainment [19], to permit adequate distribution of players per category, bandings were defined in the present study as Early ($\geq$90%) and Late (<90%), respectively.

## Experimental design

Using two separate soccer academies, 16 players from each academy (32 players in total) were randomly assigned by the primary investigator (who had no prior knowledge of the players) into teams to play 4v4 SSGs according to their bio-banding classification, with two Early teams (n = 8) and two Late teams (n = 8). The remaining players (n = 12) served as stand-by players in case of absence and injury, but these players were not used. Games were played in a mini-league format, whereby each team played the three other teams once, resulting in a total of six bio-banded SSGs (creating three game types: Early/Early, Early/Late, and Late/Late) per academy, per testing week (n = 24). The order of matches remained consistent across all the testing weeks. As per previous research designs [21], the SSGs were five-minutes in duration and were interspersed with a three-minute passive recovery period (equivalent to 60% of playing time) to limit the effects of fatigue. On completion of the bio-banding condition and following a 20-minute recovery, the same players from the bio-banding condition were then randomly re-assigned to four teams containing two Early and two Late players (fourth game type: Mixed). The adjusted teams performed another series of six SSGs per academy, per week (n = 24) that acted as a surrogate control to the bio-banded matches and was representative of current chronologically categorised practices in youth academy programmes. To coincide with normal weekly training practices, each club participated for four weeks resulting in 96 SSGs. Each week, a different pitch size was used: Week 1) small (17 x 17 m, 36.1 m$^2$ per player); week 2) medium (24 x 24 m; 72.0 m$^2$ per player); week 3) large (29.5 x 29.5 m; 108.8 m$^2$ per player); and week 4) expansive (34 x 34 m, 144.50 m$^2$ per player). All testing sessions were completed on the same day and time (evening) each week to control for normal weekly activities players may engage in prior to their soccer training. Given that 18.3 m x 23.0 m (52 m$^2$ per player) pitch sizes were deemed a limiting factor when assessing maturity-related advantages during bio-banded SSGs [21], the pitch size used within the present study were considered as an appropriate continuum of four test conditions to examine the influence of pitch size on technical characteristics of players. The present study adopted a common approach to the SSGs [21, 48] with games played outdoors on a synthetic 3G playing surface comprising no goalkeepers and two goals (2 x 1 m) placed at opposing ends of the pitch, and at the centre point between the two touchlines. Goals were only allowed to be scored in the attacking half of the pitch to encourage tactical, technical, and creative behaviours. Each game lasted five minutes with multiple balls placed around the perimeter of the pitch to increase ball-in-play time. Communication with players was limited to referee decision and score during matches to minimize the effects of verbal encouragement and feedback.

## Data collection

Technical-tactical outcome measures were collected automatically using a foot-mounted device containing an inertial measurement unit (IMU) (PlayerMaker, Florida, USA). Recording was initiated 10 minutes before activity. As per previous methods [49], players wore an IMU on each foot, with each device gathering information via a 2000˚/sec tri-axial gyroscope and a 16 g tri-axial accelerometer. The gyroscope measured angular velocity of the shank while

the accelerometer identified accelerations of the lower leg in the frontal, sagittal and transverse planes of motion. These data are processed by a machine learning algorithm designed to identify passing behaviours and categorizing both the passer and receiver using data collected at 250 Hz. Moreover, IMUs were also able to capture team characteristics including the possession percentage, the number of attempted and successful passes, the mean number of passes per possession, the mean time in possession and the mean time regaining possession. Due to a limitation within the PlayerMaker system when distinguishing between a shot and a pass [49], all shots at goal were removed from the passing network analysis, with only data that represented when one player released the ball and another received it being included. Using twelve amateur soccer players, who collectively performed 8,640 ball touches and 5,760 releases during a series of technical soccer tasks, repeated over two pre-determined distances, previous research [49] has shown the concurrent validity (agreement with video analysis) for ball touches and releases to be 95.1% and 97.6% respectively [49]. With intra-unit reliability possessing 96.9% and 95.9% agreement during soccer activity [49]. Additionally, the Game Technical Scoring Chart (GTSC) [48] was used as an alternative means to subjectively quantify the technical performance of players. Scoring was obtained from each coach's perceptions of an individual player's performance and comprised of 10 soccer-specific criteria including: assists, communication, control, cover/support, decision-making, first touch, marking, one versus one, passing and shooting. Each of these criteria was scored on a 0-to-5-point scale where: 1 – poor, 2 –below average, 3 –average, 4 –very good and 5 –excellent. A sum score to represent overall technical performance was calculated. Data collection was performed during matches by trained coaches (minimum standard of Football Association Level 2) with previous experience of using the GTSC. Coaches were allocated two players per SSG to assess. Previous research has demonstrated that the GTSC is a valid and reliable tool to quantify performance in academy soccer players [48].

## Network analysis

Passing data automatically collected from the IMUs were used to construct passing networks and calculate a series of network metrics designed to quantify performance and collective behaviours exhibited during the SSGs. Passing networks model individual players as nodes and successful passes between players as edges connecting the nodes in a weighted (edges representing more passes between two players have a higher weighting) and directed (discerning the passer from the receiver) network. Standard network metrics used to describe flow of 'information' (passes) through the network can then be used to quantify better and worse collective behaviours mediated through passing. Both macro team-based (network density and network intensity) and micro individual-based (degree centrality, closeness, betweenness and page rank) network metrics were calculated. Given the 4v4 format, a total of 12 different directed passes can be made between players, such that network density is calculated as the proportion of these potential passing options made in each game.

$$Network\ density = \frac{\sum_{i=1}^{n} \sum_{\substack{j=1 \\ j \neq i}}^{n} a_{ij}}{n(n-1)}$$

where $n$ is the number of players in the team, and $a_{ij}$ is an indicator variable that equals 1 if player $i$ passes to player $j$, and 0 otherwise. Network intensity identifies the speed at which players pass the ball while in possession and is calculated using the following equation, where $w_{ij}$ refers to the sum of all successful passes between players $i$ and $j$, and $T$ refers to the time a

team spends in possession.

$$Network\ intensity = \frac{1}{T}\sum_{i=1}^{N}\sum_{\substack{j=1 \\ j \neq i}}^{N} w_{ij}.$$

Network metrics calculated for individual players comprised a range of centrality measures that identify the importance of a player as a node within the network. Degree centrality was calculated as the total number of successful passes made and received by the player. Closeness centrality quantified the effectiveness of players to spread passing throughout the team, such that players with the highest values are those 'closest' to other players in terms of passing within the network.

$$Closeness\ centrality\ (i) = \frac{1}{\sum_i dist(i,j)}$$

where $dist(i,j)$ is the shortest path in the network between players $i$ and $j$ with higher values indicating greater closeness. Betweenness centrality identifies how often a player lies on the shortest path in the passing network from one player to another with high values denoting players that act as a conduit for moving the ball between two players.

$$Betweenness\ centrality\ (i) = \sum_{j \neq i}\sum_{k \neq i} \delta_{jk} \quad where \quad \delta_{jk}(v) = \frac{\sigma_{jk}(i)}{\sigma_{jk}}$$

where $\sigma_{jk}$ is the number of shortest paths from player $j$ to player $k$, and $\sigma_{jk}(i)$ is the number of shortest paths from player $j$ to player $k$ that travel through player $i$. The final network metric applied was the page rank algorithm that quantifies player importance using the following equation [50].

$$Page\ rank\ (i) = \frac{1-d}{N} + d\sum_{j \in M(i)} \frac{PR(j)}{C(j)}$$

where d is a damping factor, $N$ is the number of vertices in a network, $PR(j)$ is the page rank of node $j$ and $C(j)$ is the number of connecting nodes coming from node $j$. Page rank calculations represent an iterative process due to the requirement of other page rank values within the calculation.

## Statistical analysis

The effects of game type and pitch size were assessed by fitting linear models with fixed effects including pitch size (small, medium, large, expansive), game type (matched, mis-matched, mixed), and maturation (Early, Late); and player id fitted as a random effect to account for the covariance between repeated observations made on the same player. For individual based metrics, analyses were conducted by either combining (pooled) data from Early and Late players (enabling comparison of maturation) or were conducted for each group separately. Suitability of linear models were assessed and verified by visual analyses of normal distribution of random effects (using Best Linear Unbiased Predictors), and normal distribution and independence of within-group residuals and population-level residuals. Statistical inferences were made using generalized likelihood ratio tests and the appropriate standard chi-squared asymptotic reference distribution. An *a priori* alpha of 0.05 was used as a decision rule to define compatibility/ incompatibility between each hypothesis and the data (given the model used to generate each p value). We report exact p values for each comparison unless p<0.001. In addition to null

hypothesis tests (that the respective fixed effect coefficient is 0), effect magnitudes of each independent variable were quantified using Cohen's local $f^2$ for mixed-effects regression models [51] with threshold values of .02, .13 and .26 used to categorise effects as small, medium and large, respectively [52]. To indicate the precision of our effect size estimates, 95% confidence intervals (CIs) were calculated using bootstrapping comprising 1000 iterations. Statistical analyses were conducted using the lme4 package [53] in the statistical environment R (Version R-4.1.0).

## Results

A total of 3084 (55%) successful passes were made from 5605 attempts. Analysis of passing behaviours (mean (standard deviation)) showed that 29.2 (7.6) pass attempts were made per game with a success rate of 53.3 (11.8) %. Possession duration lasted 6.3 (2.3) s, with 1.4 (0.5) passes made per possession. Results were consistent with the hypothesis of no effect ($p \geq 0.438$) for either of these traditional metrics across bio-banded (Early/Early, Early/Late, Late/Late) or mixed maturity game types. However, differences and medium to large effects of pitch size were identified for mean possession ($f^2$ = 0.24 [95%CI: 0.17–0.38]; $p < 0.001$) and pass attempts ($f^2$ = 0.27 [95%CI: 0.17–0.40]; $p < 0.001$), with sequential increases in mean possession with greater pitch size (small < medium < large < expansive), coupled with sequential decreases in pass attempts (Table 1). Analysis of team-based network metrics identified similar findings, with minimal differences identified for game type ($p \geq 0.192$), but small to medium effects of pitch size with sequential decreases in network density and network intensity with greater pitch size (Table 1).

When combining data from network metrics across Early and Late players, results were consistent with the hypothesis of no effect of game type (Table 2) for degree centrality ($p = 0.201$), closeness centrality ($p = 0.086$), betweenness centrality ($p = 0.127$) and Page Rank ($p = 0.707$). In contrast, when data were analyzed across Early and Late groups separately, effects of game type were identified for Early players when placed into mixed teams. Analyses demonstrated a reduction in degree centrality ($f^2$ = 0.06 [95%CI: 0.03–0.10]; $p < 0.001$) and

**Table 1. Team-based traditional and network passing metrics and the effects of game type and pitch size.**

| Metric | Early/Early | Late/Late | Early/Late | Mixed | Game type $f^2$ [95%CI] | Pitch size $f^2$ [95%CI] |
|---|---|---|---|---|---|---|
| *Traditional metrics* | | | | | | |
| **Mean possession (s)** | 6.6 (1.5) | 5.2 (1.7) | 6.6 (2.7) | 6.3 (2.2) | $f^2$ = 0.08 [0.03–0.30] | $f^2$ = 0.24 [0.17–0.38] |
| | | | | | p = 0.438 | p < 0.001 |
| **Pass attempts** | 31.2 (7.1) | 27.8 (7.3) | 30.1 (7.9) | 28.5 (7.6) | $f^2$ = 0.00 [0.00–0.08] | $f^2$ = 0.27 [0.17–0.40] |
| | | | | | p = 0.952 | p < 0.001 |
| **Percentage completion (%)** | 58.2 (8.3) | 50.6 (11.9) | 53.9 (13.3) | 52.5 (11.1) | $f^2$ = 0.01 [0.00–0.14] | $f^2$ = 0.04 [0.02–0.12] |
| | | | | | p = 0.986 | p = 0.034 |
| **Pass per possession (%)** | 1.6 (0.4) | 1.3 (0.5) | 1.5 (0.6) | 1.4 (0.5) | $f^2$ = 0.00 [0.00–0.07] | $f^2$ = 0.01 [0.00–0.07] |
| | | | | | p = 0.830 | p = 0.314 |
| *Network metrics* | | | | | | |
| **Density** | 0.78 (0.14) | 0.60 (0.19) | 0.72 (0.20) | 0.69 (0.18) | $f^2$ = 0.04 [0.01–0.22] | $f^2$ = 0.07 [0.03–0.15] |
| | | | | | p = 0.569 | p = 0.011 |
| **Intensity** | 4.3 (0.8) | 4.2 (0.8) | 4.7 (1.8) | 5.0 (1.8) | $f^2$ = 0.03 [0.01–0.15] | $f^2$ = 0.10 [0.06–0.18] |
| | | | | | p = 0.192 | p < 0.001 |

Mean (SD). Interpretation of Cohens $f^2$ effect size can be compared to standard thresholds (small: $f^2 \geq 0.02$; medium: $f^2 \geq 0.15$; large: $f^2 \geq 0.35$). P values obtained from likelihood ratio tests comparing specific model with null model.

**Table 2. Individual-based traditional and network passing metrics and the effects of game type, pitch size and maturation.**

| Metric | Early/Early | Late/Late | Early/Late | Mixed | Effect size categories | Game type f² [95% CI] | Pitch size f² [95% CI] | Maturation f² [95% CI] |
|---|---|---|---|---|---|---|---|---|
| *Traditional metrics* | | | | | | | | |
| **GTSC Passing Score** | 3.0 (1.0) | 2.9 (0.9) | 2.9 (1.0) | 2.9 (0.9) | Pooled | f² = 0.00 [0.00–0.01] | f² = 0.02 [0.01–0.02] | f² = 0.01 [0.01–0.02] |
| | | | | | | p = 0.862 | p = 0.002 | p = 0.039 |
| | | | | | Early | f² = 0.00 [0.00–0.02] | f² = 0.01 [0.01–0.04] | - |
| | | | | | | p = 0.325 | p = 0.175 | |
| | | | | | Late | f² = 0.01 [0.00–0.03] | f² = 0.02 [0.01–0.05] | - |
| | | | | | | p = 0.221 | p = 0.066 | |
| **GTSC Sum score** | 27.1 (6.8) | 25.7 (5.5) | 25.7 (7.1) | 25.7 (6.6) | Pooled | f² = 0.00 [0.00–0.01] | f² = 0.03 [0.02–0.06] | f² = 0.04 [0.02–0.08] |
| | | | | | | p = 0.695 | p < 0.001 | p = 0.089 |
| | | | | | Early | f² = 0.01 [0.00–0.03] | f² = 0.03 [0.01–0.07] | - |
| | | | | | | p = 0.310 | p = 0.011 | |
| | | | | | Late | f² = 0.02 [0.01–0.05] | f² = 0.04 [0.02–0.09] | - |
| | | | | | | p = 0.032 | p < 0.001 | |
| *Network metrics* | | | | | | | | |
| **Degree Centrality** | 9.03 (3.65) | 8.44 (4.47) | 7.34 (4.39) | 7.79 (4.09) | Pooled | f² = 0.01 [0.00–0.02] | f² = 0.02 [0.01–0.04] | f² = 0.05 [0.01–0.06] |
| | | | | | | p = 0.201 | p = 0.001 | p = 0.003 |
| | | | | | Early | f² = 0.06 [0.03–0.10] | f² = 0.02 [0.01–0.05] | - |
| | | | | | | p < 0.001 | p = 0.061 | |
| | | | | | Late | f² = 0.01 [0.00–0.02] | f² = 0.02 [0.01–0.07] | - |
| | | | | | | p = 0.306 | p = 0.029 | |
| **Closeness Centrality** | 0.28 (0.06) | 0.26 (0.08) | 0.22 (0.08) | 0.25 (0.07) | Pooled | f² = 0.01 [0.00–0.01] | f² = 0.04 [0.03–0.07] | f² = 0.07 [0.03–0.09] |
| | | | | | | p = 0.086 | p < 0.001 | p < 0.001 |
| | | | | | Early | f² = 0.03 [0.01–0.05] | f² = 0.04 [0.03–0.09] | - |
| | | | | | | p = 0.007 | p < 0.001 | |
| | | | | | Late | f² = 0.01 [0.01–0.04] | f² = 0.05 [0.04–0.11] | - |
| | | | | | | p = 0.060 | p < 0.001 | |
| **Betweenness Centrality** | 0.67 (1.05) | 0.56 (0.87) | 0.81 (1.13) | 0.72 (1.06) | Pooled | f² = 0.01 [0.00–0.02] | f² = 0.00 [0.00–0.01] | f² = 0.00 [0.00–0.01] |
| | | | | | | p = 0.127 | p = 0.727 | p = 0.707 |
| | | | | | Early | f² = 0.03 [0.02–0.07] | f² = 0.00 [0.00–0.02] | - |
| | | | | | | p < 0.001 | p = 0.714 | |
| | | | | | Late | f² = 0.01 [0.00–0.03] | f² = 0.00 [0.00–0.02] | - |
| | | | | | | p = 0.245 | p = 0.916 | |

Mean (SD). Interpretation of Cohens f² effect size can be compared to standard thresholds (small: f²≥0.02; medium: f²≥0.15; large: f²≥0.35). P values obtained from likelihood ratio tests comparing specific model with null model.

closeness centrality (f² = 0.03 [95%CI: 0.01–0.05]; p = 0.007) and increases in betweenness centrality (f² = 0.03 [95%CI: 0.02–0.07]; p<0.001) and Page Rank (f² = 0.02 [95%CI: 0.01–0.05]; p = 0.027). Similar results were obtained for passing metrics calculated at the individual level (Table 2). When combining data from Early and Late players, results were consistent with the hypothesis of no effect of game type for GTSC passing (p = 0.862) or sum score (p = 0.695). However, small (f²≤0.03) effects (p≤0.002) were identified for changes in pitch size, with the highest values for both variables obtained during games performed on the medium sized pitch and the lowest values with the small sized pitch. Additionally, a small (f² = 0.02 [95%CI: 0.01–0.05]) effect (p = 0.032) of game type was identified for Late players GTSC sum score, with the lowest values obtained when games were played against Early players.

## Discussion

The results of this study demonstrate that bio-banding (primarily for Early players) and alteration of pitch sizes (i.e., relative pitch size) can influence the passing and tactical behaviours of academy soccer players during SSGs. The main findings of this study are fourfold: 1) Early players exhibited more effective collective behaviours than Late players; 2) bio-banding appeared to have the greatest influence on Early players whereby Early players became more integral to team dynamics when mixed with Late players; 3) bio-banding appeared to have limited effect on performance and exhibited team behaviours of Late players; and 4) smaller pitch areas tended to increase the tactical behaviours and subsequent technical performance of players.

The bio-banding process used in this study created several game types where team and opposition were comprised of matched (Early vs Early, Late vs Late) or mis-matched maturity status players (Early vs Late). Additionally, a surrogate control to represent traditional chronologically categorised (i.e., mixed maturity) methods of grouping players (Mixed vs Mixed) which is used throughout academy and grass root soccer structures was included. Previous research suggests that traditional chronologically-ordered age categorisation methods may offer sub-optimal playing environments for players situated at either extreme of the maturation continuum (i.e., late or early maturing), as these players perceive that they influence games more when competing against others of similar maturation (as prescribed when using bio-banding), particularly later-maturing player [18, 19]. However, the objective outcomes of player performance in the present study tend to suggest the findings by Cumming, Brown [19] and Bradley, Johnson [18] are intuitively exclusive to later maturing players, whereby later maturing players who are often characterised as possessing inferior anthropometric and physical fitness characteristics [10, 11, 13] become sub-consciously dependent on their early-maturing counterparts. This phenomenon is evidenced in the present study with Early-maturing players becoming more integral to passing and team dynamics (increased betweenness centrality and Page Rank) when playing in mixed-maturation teams. In contrast, the network metrics of degree and closeness centrality of Early players were lowest when playing in mixed-maturation teams, likely reflecting reduction in the overall team passing performance and collective behaviours. This perhaps suggests that academy soccer practitioners need to carefully consider the composition of teams (i.e., late: early maturing players) when implementing SSGs within non-bio-banded adolescent age groups (i.e., traditional chronological age categorised). For instance, if the objective of a SSGs training session is to enhance the technical/tactical demand of an early-maturing player, the coach may wish to consider inserting the player into a team comprised of later-maturing players.

The effects of pitch size were analysed by comparing four separate pitch sizes and dimensions categorized as small ($36.1 \text{ m}^2$ per player), medium ($72.0 \text{ m}^2$ per player), large ($108.8 \text{ m}^2$ per player), and expansive ($144.50 \text{ m}^2$ per player). A range of studies have investigated the effects of pitch area [36–39], with a recent systematic review concluding that smaller relative pitch areas ($<100 \text{ m}^2$ per player) increase the number of tactical behaviours [40]. These conclusions are consistent with the findings obtained here, with the largest effects obtained for basic team-level metrics ($f^2 = 0.24$ to $0.27$) showing shorter possession durations and increased number of passes with smaller relative pitch areas. While $0.07$ to $0.10$) indicating that the underlying structure and dynamics of passing may not be altered as much as suggested by more superficial assessments. Network density values were shown to increase with smaller relative pitch sizes reflecting greater spread of the ball among the team and linkage across different player combinations.

As expected, given the large increase in variance between team and individual-based metrics, effect sizes quantifying the influence of bio-banding processes and pitch sizes were lower for individual-based metrics. Analyses of the overall GTSC scores demonstrated no differences in maturation which is consistent with unpublished findings showing little effect of maturation on GTSC scores of academy soccer players during maturity-matched and miss-matched bio-banded SSGs [54]. These recent analyses conducted by Towlson, MacMaster [54] were restricted to a small relative pitch area (40–50 m$^2$) and it was suggested that while this may increase the number of technical actions, the type and range of actions may be restricted [55], thus limiting coaches' ability to perceive any differences in player performance. This hypothesis is supported by the results obtained in the present study, with relative pitch size having a small effect ($f^2 = 0.03$ [95%CI: 0.02–0.06]) on overall GTSC scores with the lowest values obtained during games played on the small pitch and the highest values obtained during games played on the expansive pitch. A similar ordered effect was obtained for GTSC passing scores ($f^2 = 0.02$ [95%CI: 0.01–0.03]) with the highest values associated with the expansive pitch. In direct contrast, small effects of altering pitch sizes were obtained for player-based network metrics ($f^2 = 0.00$ to 0.05) with the highest values of degree and closeness centrality obtained for the small pitch and the lowest values obtained for the expansive pitch. The contrasting results obtained for objective versus subjective measures, and basic versus more complex assessments of technical performance highlight the challenges associated with assessing the influence of soccer-based interventions. Previous research investigating SSGs and passing networks also highlighted the complexity of interventions and the potential for multiple phenomena including relative playing area, task constraints, age, position, and team composition to alter performance, collective behaviours and network properties [33, 40, 56]. In addition, maturity-matched bio-banding has been shown to have some effect on academy soccer players technical match-play characteristics during match-play formats [23, 25]. However, broader inferences regarding the efficacy of bio-banding to enhance other desirable attributes of academy players are currently difficult to establish, as trends can remain during maturity mis-matched bio-banded formats, particularly for physical characteristics [21]. That said such trends may be due to the small relative pitch size (52.6 m$^2$ per player) used [21], which likely thwart the anticipated physical and tactical advantages afforded to post-PHV players during maturity mis-matched bio-banded SSGs being exhibited. This is relevant to the present study given that the limited effect of bio-banding on tactical attributes was likely due to a higher density of players per square meter, in comparison to using a relatively larger pitch size [28], which has also been shown to constrain the type of technical actions performed by players as a function of advancing age and pitch size [55]. This was also observed in the present study, whereby greater maturity contrasts were established with advancing relative pitch size.

Although many different methods exist to estimate maturation status [6–8, 43, 46, 47], we acknowledge that the Khamis and Roche [43] method which was used to estimate the percentage of final adult height within the present study isn't without limitations. For instance, this method requires the heights of both parents for each child (not always achievable) which is often self-reported (as it was in present study) and values are sometime overestimated [42]. That said, this method is considered to be the most robust of the somatic-based, non-invasive maturity estimation methods (see Towlson, Salter [2]) as the equation encompasses a 'genetic component' (i.e., parental height) and accounts for overestimations in this measure [42]. In fact, the Khamis and Roche [43] method has been shown to possess superior prediction qualities by identifying 96% of soccer players as experiencing PHV [57], whereas original methods [7] only correctly identified 65% as experiencing PHV [57]. Previous bio-banding work has shown the Khamis and Roche [43] method to be an acceptable method for categorising

academy soccer players by maturation status, on the condition that the limitations of the method are carefully considered in relation to player characteristics being assessed [21].

## Conclusion

The results from the present study suggest that Early players performed better and exhibited more effective collective behaviours than Late players during bio-banding match-play, while Early players also appear to have a greater influence on mixed maturity match-play by becoming more integral to team dynamics. This is evidenced by increased betweenness centrality and Page Rank when playing in mixed maturation teams. However, it should also be acknowledged that additional information and potentially contrasting results may have been obtained under different formats and data analysis processes. Further research is required to identify which network metrics and analysis procedures are most informative for coaches and researchers, including the ability to predict future success in players.

### Practical applications

The findings from our study suggest two practical applications for implementing bio-banding to reduce bias which is exclusive to maturity status [15] within academy soccer players. First, (and as reported in previous studies [40]), smaller pitch sizes should be used if the training objective is to increase the frequency of technical actions, such as passing. This is evidenced in the present study with smaller playing areas enhancing passing frequency compared to larger playing areas. Second, academy practitioners should carefully consider the number of early-maturing players used within mixed maturity SSGs formats (i.e., traditional chronologically categorised formats) to create training environments that afford greater opportunity for players to achieve desired technical and tactical objectives prescribe by coaches. Although not directly examined within the present study, early evidence here suggests that SSGs teams comprised of a low early to late maturing player ratio (e.g., 1:3), using a small relative pitch size (e.g., 36.1 $m^2$ per player) may create a playing environment that could enhance the technical and tactical challenges ensued by early maturing players. This is of importance and relevance to talent development practitioners, given that early maturing players are likely characterised as possessing enhanced and temporary, maturity-related physical fitness and anthropometric characteristics [5, 9–11], which they likely (sub)consciously over-depend on (perhaps to the detriment of developing technical/tactical qualities) when contesting duals with later-maturing players during chronologically-ordered age groupings. This is somewhat supported by early-maturing players suggesting that they perceive greater physical and technical challenge when competing in maturity-matched bio-banded match-play [18]. Conversely, increasing the ratio of late to early-maturing players may reduce the physical demand imposed on later-maturing players during chronologically categorised (i.e., maturity mixed) SSGs. This is because late-maturing players have stated they perceive lower physical and technical burden when contesting maturity-matched bio-banded match-play [18]. Therefore, consideration of player maturation status during the selection of players for (non)bio-banded SSGs teams should be considered. Further research is required to establish the effect of low and high ratios of late and early-maturing players on the physical, technical, tactical and psychological responses of players during match-play.

## Acknowledgments

The authors would like to thank all players and coaching staff at the soccer club for their cooperation and assistance during this study.

## Author Contributions

**Conceptualization:** Christopher Towlson, Grant Abt, Sean Cumming.

**Data curation:** Christopher Towlson, Steve Barrett, Alex Lowthorpe, Martin Corsie, Paul Swinton.

**Formal analysis:** Martin Corsie, Paul Swinton.

**Investigation:** Christopher Towlson, Grant Abt, Ally Hamilton, Alex Lowthorpe, Bruno Goncalves, Martin Corsie.

**Methodology:** Christopher Towlson, Grant Abt, Steve Barrett, Sean Cumming, Frances Hunter, Ally Hamilton, Alex Lowthorpe, Bruno Goncalves, Martin Corsie, Paul Swinton.

**Project administration:** Christopher Towlson, Frances Hunter.

**Resources:** Steve Barrett, Frances Hunter, Ally Hamilton, Paul Swinton.

**Software:** Steve Barrett, Paul Swinton.

**Supervision:** Christopher Towlson, Grant Abt, Steve Barrett.

**Visualization:** Paul Swinton.

**Writing – original draft:** Christopher Towlson, Grant Abt, Sean Cumming, Frances Hunter, Ally Hamilton, Alex Lowthorpe, Bruno Goncalves, Martin Corsie, Paul Swinton.

**Writing – review & editing:** Christopher Towlson, Grant Abt, Sean Cumming, Paul Swinton.

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
