## [Decision Letter · Decision Letter 0]

23 Aug 2021

PONE-D-21-21543

The effect of bio-banding on academy soccer player passing networks: Implications for relative pitch size

PLOS ONE

Dear Dr. Towlson,

Thank you for submitting your manuscript to PLOS ONE. After careful consideration, we feel that it has merit but does not fully meet PLOS ONE’s publication criteria as it currently stands. Therefore, we invite you to submit a revised version of the manuscript that addresses the points raised during the review process.

ACADEMIC EDITOR:

Dear Authors, 

two experts in the field revised your manuscript founding some issues yous should address.

We look forward to receiving your revised manuscript.

Kind regards,

Emiliano Cè

Academic Editor

PLOS ONE

Journal Requirements:

Reviewers' comments:

Reviewer's Responses to Questions

**Comments to the Author**

1. Is the manuscript technically sound, and do the data support the conclusions?

Reviewer #1: Partly

Reviewer #2: Partly

2. Has the statistical analysis been performed appropriately and rigorously? 

Reviewer #1: Yes

Reviewer #2: Yes

3. Have the authors made all data underlying the findings in their manuscript fully available?

Reviewer #1: Yes

Reviewer #2: Yes

4. Is the manuscript presented in an intelligible fashion and written in standard English?

Reviewer #1: Yes

Reviewer #2: Yes

5. Review Comments to the Author

Reviewer #1: Dear Editor in-Chief,

I would like to thanks for the opportunity to revise this article entitled “The effect of bio-banding on academy soccer player passing networks: implications of relative pitch size”.

In academy players, the use of bio-bending is a very interesting aspect to deliver a very individualized training prescription. However, it is not easy to find the right way to apply all this information into practice due to a mandatory chronological classification for championship participation (U19, U18, U17, etc). As such, future perspectives are required and I would like to thanks the authors for their contribution.

The article is well written and easy to understand. However, honestly, I find the practical applications quite weak. Firstly, the results about the effects of pitch size on technical aspects (increase of technical actions) are well acknowledged. Secondly, the use of bio-bending analysis integrated with technical-tactical analysis require a lot of work to be well determined within training routine. Therefore, I would to highlighted the great work performed by the Authors but I would to invite them to reworded and rethink the use of these information as ‘game-changer information’. Could these information and analysis change the approach to selection and increase the quality of training? Could the use of bio-bending analysis integrated to SSGs provide the opportunity to increase the training quality and performance development of the youth players? In my opinion the work required to create these specific analyses inside elite academy training routine should be more highlighted. I would kindly ask to the Authors to revise their paper (discussion, conclusion and practical applications) considering the aforementioned questions, please. As practitioners, reading this paper I need to be convinced that is mandatory required this analysis and that I need to involve practitioners daily to analyze this information in practice to increase the quality of performance development in elite academies (I suggest to mainly consider passing metric network than pitch-size).

Introduction:

Line 70: please provide some previous findings about how the readers could apply bio-banding information to increase the quality of their work.

Line 76 to 89: this paragraph is not clear. Please, try to simplify

Line 86-89: If the authors would to introduce the effects of SSG pitch size I believe that could be interesting to briefly speak about effects of pitch size on tactical, physical and physiological aspect as previously stated for example in Olthof et al JSS 2018, Riboli et al plos one 2020, Castagna et al IJSPP 2019, etc..

Materials and methods

Line 148: In my opinion would be useful to provide here how long each SSG lasted (as reported at line 157)

Line 190: in my opinion could be interesting to provide a representative figure (graphical representation) of network analysis to support figure and equations (are these latter mandatories required?)

Discussion:

I believe that the discussion and especially conclusion and practical applications should be more based on the application of these results and it should convince the readers about the effective needs to increase the integration of bio-banding analysis with network analysis into daily practice, as previously suggested. Please, provide why this information could be ‘game-changer’ information and affect training prescriptions on daily base. The Authors could convince readers that is very important to integrate all these analysis into daily practice, not only suggest to play with three late-maturing and one early-maturing player. This paper could have great new practical applications opening to great future perspective for training prescriptions but the author should suggest these solutions, please.

Reviewer #2: I would like to thank the editors for the opportunity of reviewing this manuscript. This work aims to examine the effect of bio-banding players on passing networks creating during 4-a-side games. I find this work very interesting and relevant. I think it is, in general, well written. However, I think there are several weaknesses (mainly in methods) that need to be addressed before considering this work for publication. I hope my comments help to improve the quality of this manuscript. I am also happy to discuss my comments with the authors in case they disagree. Please, see my comments in the attached document.

6. PLOS authors have the option to publish the peer review history of their article (what does this mean?). If published, this will include your full peer review and any attached files.

Reviewer #1: No

Reviewer #2: **Yes: **Jorge López-Fernández

---

## [Author Response · Author response to Decision Letter 0]

19 Oct 2021

We sincerley thank the reviewers and editor for their time. 

Reviewers' comments:

Reviewer's Responses to Questions

Reviewer 1

I would like to thank the editors for the opportunity of reviewing this manuscript. This work aims to examine the effect of bio-banding players on passing networks creating during 4-a-side games. I find this work very interesting and relevant. I think it is, in general, well written. However, I think there are several weaknesses (mainly in methods) that need to be addressed before considering this work for publication. I hope my comments help to improve the quality of this manuscript. I am also happy to discuss my comments with the authors in case they disagree. 

We thank the reviewer in advance for their positive words and constructive feedback. 

ABSTRACT

I suggest the authors go straight to the study’s objective, which in fact are three, not two, as stated in the abstract. 

Methods: I think they are not clear and need to be improved. It is not clear anything related to pitch size nor a subjective-coach-based scoring system.

Results: I think they are described chaotically and are not easy to understand by the reader

Conclusion: I suggest the authors go straight to the point, so they might save some words for describing the methods more accurately.

KEYWORD: avoid using the same words as in the title. For instance, soccer might be replaced by football.

We thank the reviewer for their comments. We agree with most of the points raised regarding the abstract. However, we feel the existing opening lines of the abstract do in fact outline the aims/objectives of the manuscript, and we do not feel it necessary to alter this. That said, we have overhauled the rest of the abstract based on your comments and we feel that that your feedback has improved this part of the manuscript.

Abstract

The primary aims of this study were to examine the effects of bio-banding players on passing networks created during 4v4 small-sided games (SSGs), while also examining the interaction of pitch size using passing network analysis compared to a coach-based scoring system of player performance. Using a repeated measures design, 32 players from two English Championship soccer clubs contested mixed maturity and bio-banded SSGs. Each week, a different pitch size was used: Week 1) small (36.1 m2 per player); week 2) medium (72.0 m2 per player); week 3) large (108.8 m2 per player); and week 4) expansive (144.50 m2 per player). All players contested 12 maturity (mis)matched and 12 mixed maturity SSGs. Technical-tactical outcome measures were collected automatically using a foot-mounted device containing an inertial measurement unit (IMU) and the Game Technical Scoring Chart (GTSC) was used to subjectively quantify the technical performance of players. Passing data collected from the IMUs were used to construct passing networks. Mixed effect models were used with statistical inferences made using generalized likelihood ratio tests, accompanied by Cohen's local f2 to quantify the effect magnitude of each independent variable (game type, pitch size and maturation). Consistent trends were identified with mean values for all passing network and coach-based scoring metrics indicating better performance and more effective collective behaviours for early compared with late maturation players. Network metrics established differences (f2 = 0.00 to 0.05) primarily for early maturation players indicating that they became more integral to passing and team dynamics when playing in a mixed-maturation team. However, coach-based scoring was unable to identify differences across bio-banding game types (f2 = 0.00 to 0.02). Pitch size had the largest effect on metrics captured at the team level (f2 = 0.24 to 0.27) with smaller pitch areas leading to increased technical actions. The results of this study suggest that the use of passing networks may provide additional insight into the effects of interventions such as bio-banding and that the number of early-maturing players should be considered when using mixed-maturity playing formats to help to minimize late-maturing players over-relying on their early-maturing counterparts during match-play.

Keywords: Football, small-sided games, talent development, talent identification, bio-banding, maturation.

 

INTRODUCCTION

Comment 1: I like the way the introduction of the paper has been written. However, I think it might be appropriate to provide further comments on the need of evaluating pitch size. Moreover, the statement in lines 116-117 requires a reference. 

We thank the reviewer for their comment here. We have revised the section.

“Therefore, the primary aim of this study was to investigate the effects of bio-banding players on passing networks created during 4v4 SSGs. In addition, as pitch size can influence tactical actions of players [36-39], with smaller pitch areas (<100 m2 per player) being shown to increase the number of such actions [40] and larger pitches eliciting greater physical demands and more opportunity for players to record higher running speeds [41], the interaction of pitch size was also investigated. Finally, given that coach observations are often the first point of player evaluation (i.e., scouts), it is important to establish if coach observations provide a suitable assessment of players technical and tactical actions during match-play. Therefore, network analysis measures were compared to a subjective coach-based scoring system of player performance during bio-banded match-play.

Comment 2: I also think it might be appropriate to provide further information to justify why the comparison between network analysis and subjective-coach-based scoring system is required. 

Thank you for your comments here. We agree and have made the following revisions.

 “In addition, as pitch size can influence tactical actions (31-34), with smaller pitch areas (<100 m2 per player) being shown to increase the number of technical actions (35) and larger pitches eliciting greater physical demands and more opportunity for players to record higher running speeds (36), the interaction of pitch size was also investigated. Finally, given that coach observations are often the first point of player evaluation (i.e., scouts etc), it is important to establish if coach observations provide a suitable assessment of players technical and tactical actions during match-play. Therefore, network analysis measures will be compared to subjective coach-based scoring system of player performance during bio-banded match-play.”

METHODS 

Participants

I would like to know why maturation status was only measured using height. Why was not participants’ sexual development evaluated by other instruments like the Tanner Scale?

Thank you for your comment. Pleased find a revised statement 

 “Anthropometric data were combined with age and self-reported parental height adjusted for over-estimation [42], with player estimated percentage of parental adult height (PAH%) calculated using the Khamis and Roche [43] method. The Khamis and Roche [43] method is commonly used within academy soccer programmes [44], often as a surrogate for more invasive measures of biological maturation (e.g., stage of pubic hair development [45, 46] and skeletal age [47]). Although we recognise that PHV onsets at approximately 86% of estimated adult stature attainment [19], to permit adequate distribution of players per category, bandings were defined in the present study as Early (≥90%) and Late (<90%), respectively.”

Experimental design

I do not think this is clear. I read this section several times and I have still doubts. I encourage the authors to rewrite this section to facilitate understanding. 

Lines 145-146: According to this statement, only 2 Early teams and 2 Late teams were created for this study (in total 16 players participated). This is not clear, as 44 players were recruited in total (24 from team 1, and 20 from team 2). Also, it is not clear if players from both teams were mixed or not. it is not clear either who created each 4-a-side team. Were they created randomly only considering the bio-banding score? Were participants gathered into their team according to coach criteria? Finally, if only 16 players per team participated what happened with the remaining players? If all players participated, can this fact bias the outcomes?

Thank you for raising this concern. We have made the below amendments to clarify the sample sizes. 

“Players were initially over-recruited (to permit an adequate numbers of players per maturity banding to be identified, while accounting for attrition) from two English Championship clubs, including one in category 1 (n = 24) and one in category 2 (n = 20).”

“Using two separate soccer academies, 16 players from each academy (32 players in total) were randomly assigned by the primary investigator (who had no prior knowledge of the players) into teams to play 4v4 SSGs according to their bio-banding classification, with two Early teams (n = 8) and two Late teams (n = 8). The remaining players (n = 12) served as stand-by players in case of absence and injury, but these players were not used.”

Lines 146-147: I know the term “round robin”, and to my knowledge, it just means all teams play each other once. I might be wrong as I am not English, but I encourage the authors to be specific and clearly state that each team played against the other three teams a total of two times. Otherwise, I cannot understand how each team played 6 games.

Thank you for your comment here. We have made the following revision.

“Games were played in a mini-league format, whereby each team played the three other teams once, resulting in a total of six bio-banded SSGs (creating three game types: Early/Early, Early/Late, and Late/Late) per academy, per testing week (n = 24).”

Lines 147-148: I understand the three game’s types but, I think it should be appropriate to state that Early vs Early and Late vs Late were played two times. On the contrary, Early vs Late was played four times. 146-148: Which were the matches order? were they always the same? I think it might bias the outcomes.

We thank the reviewer for their comment here. Given that there were 2 late and two early teams who played each other once per week, there were 6 bio-banded fixtures per week (four weeks in total), per academy (two academies)

We have revised the below in the hope that this is made much clearer. 

 “Using two separate soccer academies, 16 players from each academy (32 players in total) were randomly assigned by the primary investigator (who had no prior knowledge of the players) into teams to play 4v4 SSGs according to their bio-banding classification, with two Early teams (n = 8) and two Late teams (n = 8). The remaining players (n = 12) served as stand-by players in case of absence and injury, but these players were not used. Games were played in a mini-league format, whereby each team played the three other teams once, resulting in a total of six bio-banded SSGs (creating three game types: Early/Early, Early/Late, and Late/Late) per academy, per testing week (n = 24).”

Line 148-149: Why were 3 minutes of passive recovery set? I am not sure it is enough to limit fatigue. Is there a reference that can be used to justify this? Please, notice that if 6 matches were played in total players played for 30 minutes. Fatigue might exist in last two or three games.

Thank you for this feedback. We have inserted additional information for context. 

 “As per previous research designs [21], the SSGs were five-minutes in duration and were interspersed with a three-minute passive recovery period (equivalent to 60% of playing time) to limit the effects of fatigue.”

Lines 149-150: It is not clear at all the players re-assignation. I think it might be easier to understand if the authors explain at the beginning of this section that two different tests were conducted. Firstly, players were gathered into a team according to their bio-banding score, while secondly games were conducted gathering two early players with two late players in each team. 

We thank the reviewer for raising this and we have revised the statement below for clarity.

 “On completion of the bio-banding condition and following a 20-minute recovery, the same players from the bio-banding condition were then randomly re-assigned to four teams containing two Early and two Late players (fourth game type: Mixed). The adjusted teams performed another series of six SSGs per academy, per week (n =24) that acted as a surrogate control to the bio-banded matches and was representative of current chronologically categorised practices in youth academy programmes.”

Lines 150-152: Why were 6 matches played in “mixed”? This fact increases the number of observations of mixed games while the other three games observations are much lower. Also, I guess chronological do not always meet the criteria of 2 Early players and 2 Late players. So, might this fact bias the outcomes?

There were in fact 6 fixtures performed. 

152-154: How far was each goal from each other and from touch line?

We thank the reviewer for their comment and have added some extra detail for context.

 “The present study adopted a common approach to the SSGs [21, 48] with games played outdoors on a synthetic 3G playing surface comprising no goalkeepers and two goals (2 x 1 m) placed at opposing ends of the pitch, and at the centre point between the two touchlines. Goals were only allowed to be scored in the attacking half of the pitch to encourage tactical, technical, and creative behaviours. Each game lasted five minutes with multiple balls placed around the perimeter of the pitch to increase ball-in-play time. Communication with players was limited to referee decision and score during matches to minimize the effects of verbal encouragement and feedback."

Line 158-162: I think it should be appropriate to inform at the beginning of this section that in total four different 4-a-side games were played. There is a lot of information here. Maybe it might be useful to use a Table or Figure to display part of this information in a clearer way.

Line 158: Why was only one test-day completed per week?

We thank the reviewer for their comment here, we have added some extra content to add clarity

 “To coincide with each clubs normal weekly training practices, each club participated for 4 weeks resulting in 96 SSG`s.”

Additional comments: I think that further information is needed here. Was there some kind of control to avoid participants to be overload or fatigued before each test-day? Was diet controlled? When was the test-day conducted? Did participants tested for 24, 48 or 72-h before each test-day and did not engage in vigorous physical activity within this period? I think this is important as might bias the outcomes.

No we didn’t necessarily control for such considerations. That said, all testing sessions were completed on the same day and time (evening) each week in an attempt to control for normal weekly activities players may engage in prior to their soccer training. This information has now been inserted to add more context. 

Data collection

Lines 169-179: I know PlayerMaker devices, but I am not too familiarised with them. According to the authors, this device can identify tactical information related to passing. I think I know how it works, but there are some important information missing: 

1. how does the system distinguish between a failure pass or a shot to goal?

This is a limitation of the system, and it can only distinguish when a player releases the ball and receives the ball. Within the software, to calculate the passing networks. In order to clarify this, please see the revision referenced below.

2. How accurate is PlayerMaker in collecting passing-related information? 

Playermaker/ foot-mounted IMU’s have been utilised to detect when a player receives and releases a ball with 0.96 to 1.00 ICC’s in comparison to traditional performance analysis methods (Marris et al., 2021). In order to clarify this, please see the revision referenced below.

3. Is this system validated for this study’s purpose?

Yes, the system has been validated for when a player receives and releases the ball through identification (Marris et al., 2021). In order to clarify this, please see the revision referenced below.

4. Has been this system used in previous studies to analyse passing behaviour of soccer players?

This is the first study to our knowledge that has utilised foot mounted IMU’s to detect passing networks. Studies have been performed to detect when players receive and release the ball (Marris et al., 2021), but no specifics regarding passing networks. In order to clarify this, please see the revision referenced below.

5. Is this system also able to identify the distance of each player from each other?

No, the system is unable to do this as it only contains accelerometer/ gyroscopes, and no positional data is included within the system. In order to clarify this, please see the revision referenced below.

We thank the reviewer for their comments here. We have made the following revision.

 “Due to a limitation within the PlayerMaker system when distinguishing between a shot and a pass [49], all shots at goal were removed from the passing network analysis, with only data that represented when one player released the ball and another received it being included. Using twelve amateur soccer players, who collectively performed 8,640 ball touches and 5,760 releases during a series of technical soccer tasks, repeated over two pre-determined distances, previous research [49] has shown the concurrent validity (agreement with video analysis) for ball touches and releases to be 95.1% and 97.6% respectively [49]. With intra-unit reliability possessing 96.9% and 95.9% agreement during soccer activity [49].”

Line 180-186: I also think there is quite a lot of information missing here:

1. How many coaches used the Game Technical Scoring Chart? 

2. were they familiarised with this instrument? 

3. Line 186: were some coaches only assessing one player? 

4. Also, I wonder why this tool was selected given the fact it measures other behaviours besides passing. 

5. Many readers will not be familiarised with the F.A. level so I think it should be clarified what means to be F.A. Level 1 coach. 

Again, we thank you for insightful comments and we have provided some extra detail to strengthen the clarity of the section. 

 “A sum score to represent overall technical performance was calculated. Data collection was performed during matches by trained coaches (minimum standard of Football Association Level 2) with previous experience of using the GTSC. Coaches were allocated two players per SSG to assess. Previous research has demonstrated that the GTSC is a valid and reliable tool to quantify performance in academy soccer players [48]”

Results:

Line 247: This first sentence is repeated. So, it should be removed from here or from methods. 

Thank you for pointing this out. We have removed the opening sentence. 

Line 249: What does the number in brackets mean? Standard deviation? This fact is described later in the line so it should be mentioned earlier. Also, I think it might be appropriate to insert the symbol ±

Thank you for pointing this out. We have made it clear in the first sentence that this is the standard deviation and have included the symbol as suggested. 

Line 251: I think it might be good to remember that “game type” mean the (Early/Early, Early/Late, Late/Late and mixed). 

Thank you for raising this point. We have adjusted this section accordingly.

 “No differences (p≥0.438) were identified for either of these traditional metrics across different across bio-banded (Early/Early, Early/Late, Late/Late) or mixed maturity game types.”

Lines 260-273 and Table 2: in order to be consistent with the objectives and to facilitate the reading, I recommend the authors to first talk about the data from the network passing analysis and then data from Game Technical Scoring Chart. I think this is the order that the reader expects finding the information.

Thank you for this point and we agree that this change would benefit the flow of the manuscript and have altered accordingly. 

“When combining data from network metrics across Early and Late players, results were consistent with the hypothesis of no effect of game type (Table 2) for degree centrality (p=0.201), closeness centrality (p=0.086), betweenness centrality (p=0.127) and Page Rank (p=0.707). In contrast, when data were analyzed across Early and Late groups separately, effects of game type were identified for Early players when placed into mixed teams. Analyses demonstrated a reduction in degree centrality (f2=0.06 [95%CI: 0.03-0.10]; p<0.001) and closeness centrality (f2=0.03 [95%CI: 0.01-0.05]; p=0.007) and increases in betweenness centrality (f2=0.03 [95%CI: 0.02-0.07]; p<0.001) and Page Rank (f2=0.02 [95%CI: 0.01-0.05]; p=0.027). Similar results were obtained for passing metrics calculated at the individual level (Table2). When combining data from Early and Late players, results were consistent with the hypothesis of no effect of game type for GTSC passing (p=0.862) or sum score (p=0.695). However, small (f2≤0.03) effects (p≤0.002) were identified for changes in pitch size, with the highest values for both variables obtained during games performed on the medium sized pitch and the lowest values with the small sized pitch. Additionally, a small (f2=0.02 [95%CI: 0.01-0.05]) effect (p=0.032) of game type was identified for Late players GTSC sum score, with the lowest values obtained when games were played against Early players.”

Pooled: Why “pooled” is used to mean mixed team instead of “mixed” as described in methods? This comment is also for the text. 

The term pooled is used when it combines both Early and Late data (e.g. Early vs. Early, Early vs Late and Mixed). We have made this clearer in both the statistics section where we explain the use of the term and by also making it clear that it is combining data in the results section. 

 “For the mixed maturity SSG data comprising Early and Late players across all games, no difference of game type was identified for GTSC passing (p=0.862) or sum score (p=0.695).”

DISCUSSION

Lines 277-279: I am not that sure about this statement. Bio-banding effect was only found in early players when moving from Early team to mixed-team. So, I think this first statement is not totally right. Actually, there is not comparison between early players and late players, comparisons described in results the comparison described are among game type and pitch size. Table two provide comparison between early and late players, but I am not sure what is being compared as this information is not explain in statistical analysis section. In any case, I do not see a direct comparison between early players and late players for the different studied variables. 

We have added in the opening sentence that the effects were noted for Early players. We have also added information to the statistical analysis section to make it clearer what models were completed by identifying the levels of the different fixed effects. As you have identified, the only comparisons that could be made of early vs late were for individual-based metrics that pooled data from both Early and Late players (this is now made clear in the stats section). 

Line 279 (finding 1): What is better and more effective collective behaviour? Higher number of passing? higher passing accurate? I do not think “better” is the right word, I would recommend use a different word. On the other hand, earlier players only showed a reduction in degree centrality, closeness centrality and increases in betweenness centrality when comparing their performance in bio-banding matches and mixed matches. The authors do not identify differences between early and late players in results. Accordingly, I do not think this statement is too accurate. 

We agree that better is not the correct summary and have removed this statement. Our finding is based on the consistent trend in the data regarding higher values for Early/Early vs Late/Late for possession, pass attempts, percentage completion, pass per possession, density, intensity, GTSC Sum score, degree centrality, closeness centrality and page rank. These were tested specifically and found to be significant for GTSC passing, degree centrality, closeness centrality and page rank. Collectively, we feel that the data provides evidence of greater performance/behaviours of Early players and matches with the observations of coaches and previous research. 

Line 283: I do not think pass attempts is a technical action, but a tactical behaviour. Network density and network intensity are not technical action either so, I do not think this statement is right. 

We agree with your comment here and have altered the statement to acknowledge you point. 

 “4) smaller pitch areas tended to increase the tactical behaviours and subsequent technical performance of players.” 

Lines 284-298: I do not think this present study can be compared with the work of Cumming and Brown (18) or Bradley and Johnson (17) as these studies analysed players’ perception while the present work is analysing passing behaviour. 

We agree that Sean Cummings and Ben Bradleys fine work is focused on perceptual measures. However, the objective data we provide here perhaps provides justification for informed speculation (as we obviously didn’t measure perceptions also) to suggest the findings by Sean and Ben are intuitively exclusive to later maturing players, whereby later maturing players who are often characterised as possessing inferior anthropometric and physical fitness characteristics become sub-consciously dependent on their early-maturing counterparts. We therefore feel that this point is relevant and may help to justify/inform future research questions. 

Line 301-310: I agree, but passing behaviour is not a technical action but a tactical response to the game demands. Passing is a tactical action not technical. Technical is kicking the ball, controlling the ball, etc.

We agree and have altered the below statement accordingly.

 “A range of studies have investigated the effects of pitch area (31-34), with a recent systematic review concluding that smaller pitch areas (<100 m2 per player) increase the number of tactical behaviours (35). These conclusions are consistent with the findings obtained here, with the largest effects obtained for basic team-level metrics (f2 = 0.24 to 0.27) showing shorter possession durations and increased number of passes with smaller pitch areas.”

Additional comment: I think the authors must disclose the main limitation of their study. 

 “Although many different methods exist to estimate maturation status (6-8, 38, 41, 42), we acknowledge that the Khamis and Roche (38) method which was used to estimate the percentage of final adult height within the present study isn’t without limitations. For instance, this method requires the heights of both parents for each child which is often self-reported (as it was in present study) and values are sometime overestimated (37). That said, this method is considered to be the most robust of the somatic-based, non-invasive maturity estimation methods (see Towlson, Salter (2)) as the equation encompasses a ‘genetic component’ (i.e., parental height) and accounts for overestimations in this measure (37). In fact, the Khamis-Roche(38) method has been shown to possess superior prediction qualities by identifying 96% of soccer players as experiencing peak height velocity (52), whereas original methods (7) correctly identified 65% as experiencing PHV(52). Previous bio-banding work has shown the Khamis and Roche (38) method to being an acceptable method for categorising academy soccer players by maturity status, on the condition that the limitations of the method are carefully considered in relation to player characteristics being assessed (20).”

CONCLUSIONS

Paragraph 1: The authors should go straight to the point and display the main conclusion of their study. Most of the information in this paragraph does not belong to conclusion

We acknowldeg your point and have removed the statement suggesting this. The section now reads as follows

 “The results from the present study suggest that Early players performed better and exhibited more effective collective behaviours than Late during bio-banding match-play, while Early players also appear to have a greater influence on mixed maturity match-play by becoming more integral to team dynamics. This is evidenced by increased betweenness centrality and Page Rank when playing in mixed maturation teams. However, it should also be acknowledged that additional information and potentially contrasting results may have been obtained under different formats and data analysis processes. Further research is required to identify which network metrics and analysis procedures are most informative for coaches and researchers, including the ability to predict future success in players.”

Paragraph 2: I recommend the authors to add an additional subheading for this information named “Practical application” given this information does not belong to conclusion. 

Agreed. We have added this sub header.

Lines 365—370: I am not sure this can be postulated. The authors did not investigate the behaviour of players when a team is made by 3 late players and 1 early player. Therefore, this assumption is not supported in this work. 

We acknowldeg your point and have removed the statement suggesting this. The section now reads as follows

Reviewer 2

I would like to thanks for the opportunity to revise this article entitled “The effect of bio-banding on academy soccer player passing networks: implications of relative pitch size”.

In academy players, the use of bio-bending is a very interesting aspect to deliver a very individualized training prescription. However, it is not easy to find the right way to apply all this information into practice due to a mandatory chronological classification for championship participation (U19, U18, U17, etc). As such, future perspectives are required and I would like to thanks the authors for their contribution.

The article is well written and easy to understand. However, honestly, I find the practical applications quite weak. Firstly, the results about the effects of pitch size on technical aspects (increase of technical actions) are well acknowledged. Secondly, the use of bio-bending analysis integrated with technical-tactical analysis require a lot of work to be well determined within training routine. Therefore, I would to highlighted the great work performed by the Authors but I would to invite them to reworded and rethink the use of these information as ‘game-changer information’. 

Could these information and analysis change the approach to selection and increase the quality of training? 

Could the use of bio-bending analysis integrated to SSGs provide the opportunity to increase the training quality and performance development of the youth players? 

In my opinion the work required to create these specific analyses inside elite academy training routine should be more highlighted. I would kindly ask to the Authors to revise their paper (discussion, conclusion and practical applications) considering the aforementioned questions, please. 

As practitioners, reading this paper I need to be convinced that is mandatory required this analysis and that I need to involve practitioners daily to analyze this information in practice to increase the quality of performance development in elite academies (I suggest to mainly consider passing metric network than pitch-size). 

Introduction:

Line 70: please provide some previous findings about how the readers could apply bio-banding information to increase the quality of their work.

We thank the reviewer for their comment. However, we feel that the use/application/findings of published bio-banding work is introduced within the following paragraph.

“To control for the confounding influence of maturation alone [15] during talent identification and development, researchers and practitioners have grouped players according to maturation status (typically referred to as ‘bio-banding’ [16, 17]) to create homogenous groups of players who are primarily ‘matched’ for maturity-related anthropometric characteristics. However, despite players and key stakeholders valuing the approach [18-20], there is limited applied soccer-based research to support its efficacy [21-25]. That said, Abbott, Williams (23) reported that matching players for maturity status (i.e., late maturing vs late maturing, early maturing vs early maturing) during match-play may control maturity-related differences in physical match-activity profiles, while altering the technical demands (e.g., shots, dribbles, tackles etc). In addition, Towlson, MacMaster (21) have stated that mis-matching players (i.e., late maturing vs early maturing) during bio-banded formats may enhance the identification of desirable psychological characteristics of pre-PHV academy soccer players [21]. However, they also suggested that the small, relative size of the single pitch used in the study may have limited the expression of other maturity-related match-play characteristics. This is important to practitioners responsible for identifying talented soccer players given that contextual match factors such as larger relative pitch size likely afford earlier-maturing players the opportunity to apply tactical superiority due to their transient anthropometric, physical fitness and decision-making characteristics [10, 11, 13, 26]. Such considerations are particularly important when considering player performance, as physical performance is position specific [27] and can significantly increase on a large pitch, with inter-team and intra-team distances becoming significantly larger, subsequently increasing within-team tactical variability (i.e., intra-team distances) [28].”

Line 76 to 89: this paragraph is not clear. Please, try to simplify 

We thank the reviewer for their comment here and have added some extra context to better explain the key terms of maturity matched and mis-matched. We hope this helps.

 “However, despite players and key stakeholders valuing the approach [18-20], there is limited applied soccer-based research to support its efficacy [21-25]. That said, Abbott, Williams (23) reported that matching players for maturity status (i.e., late maturing vs late maturing, early maturing vs early maturing) during match-play may control maturity-related differences in physical match-activity profiles, while altering the technical demands (e.g., shots, dribbles, tackles etc). In addition, Towlson, MacMaster (21) have stated that mis-matching players (i.e., late maturing vs early maturing) during bio-banded formats may enhance the identification of desirable psychological characteristics of pre-PHV academy soccer players [21]. However, they also suggested that the small, relative size of the single pitch used in the study may have limited the expression of other maturity-related match-play characteristics. This is important to practitioners responsible for identifying talented soccer players given that contextual match factors such as larger relative pitch size likely afford earlier-maturing players the opportunity to apply tactical superiority due to their transient anthropometric, physical fitness and decision-making characteristics [10, 11, 13, 26]. Such considerations are particularly important when considering player performance, as physical performance is position specific [27] and can significantly increase on a large pitch, with inter-team and intra-team distances becoming significantly larger, subsequently increasing within-team tactical variability (i.e., intra-team distances) [28].”

Line 86-89: If the authors would to introduce the effects of SSG pitch size I believe that could be interesting to briefly speak about effects of pitch size on tactical, physical and physiological aspect as previously stated for example in Olthof et al JSS 2018, Riboli et al plos one 2020, Castagna et al IJSPP 2019, etc.

We agree that brief discussion of the points you raise would add valuable context. Please find the revised section below.

 “This is important to practitioners responsible for identifying talented soccer players given that contextual match factors such as larger relative pitch size likely afford earlier-maturing players the opportunity to apply tactical superiority due to their transient anthropometric, physical fitness and decision-making characteristics (10, 11, 13, 24). Such considerations are particularly important when considering player performance, as physical performance is position specific (25) and can significantly increase on a large pitch, with inter-team and intra-team distances becoming significantly larger, subsequently increasing the tactical variability (i.e., intra-team distances) (26)”

Materials and methods

Line 148: In my opinion would be useful to provide here how long each SSG lasted (as reported at line 157)

Thank you for your comment. We have revised the section accordingly.

 “The order of matches remained consistent across all the testing weeks. As per previous research designs (21), the SSG`s were 5 minutes in duration and were interspersed with a 3-minute passive recovery period (equivalent to 60% of actual playing time) to limit the effects of fatigue.” 

Line 190: in my opinion could be interesting to provide a representative figure (graphical representation) of network analysis to support figure and equations (are these latter mandatories required?)

We agree that figures can be instructive, in particular they work well with spatio-temporal metrics. However, with many of the network metrics including page rank they don’t tend to translate as well to figures regarding players and are more related to the structure of the passing matrix. We agree that equations are not internalised by all readers. However, in a recent review article that we have under review, we highlighted that there are often multiple approaches that can be used to obtain the same metric (with both spatio-temporal and network) and so for best practice and transparency, equations are best presented. 

Discussion:

I believe that the discussion and especially conclusion and practical applications should be more based on the application of these results and it should convince the readers about the effective needs to increase the integration of bio-banding analysis with network analysis into daily practice, as previously suggested. Please, provide why this information could be ‘game-changer’ information and affect training prescriptions on daily base. The Authors could convince readers that is very important to integrate all these analysis into daily practice, not only suggest to play with three late-maturing and one early-maturing player. This paper could have great new practical applications opening to great future perspective for training prescriptions but the author should suggest these solutions, please. 

Thank you for your insightful feedback. We consider it important that our research has practical application to practitioners. We have therefore rewritten this section accordingly.

Practical applications 

 “The findings from our study suggest two practical applications for implementing bio-banding to reduce bias which is exclusive to maturity status (15) within academy soccer players. First, smaller pitch sizes should be used if the training objective is to increase the frequency of technical actions, such as passing. This is evidenced in the present study with smaller playing areas enhancing passing frequency compared to larger playing areas. Second, academy practitioners should carefully consider the number of early-maturing players within mixed maturity SSG formats to create more optimum training environments that afford greater opportunity for players to achieve desired technical and tactical objectives prescribe by coaches. Although not directly examined within the present study, evidence here suggests that SSG teams comprised of a low early to late maturing player ratio (e.g., 1:3), using a small relative pitch size (e.g., 36.1 m2 per player) may create a playing environment which could enhance the technical and tactical loads ensued my early maturing players. This is of importance and relevance to talent development practitioners, given that early maturing players are likely characterised as possessing enhanced and temporary, maturity-related physical fitness and anthropometric characteristics (5, 9-11) which they likely (sub)consciously over-depend on (perhaps to the detriment of developing technical/tactical qualities) when contesting duals with later maturing players during chronologically-ordered age groupings. This somewhat evidenced by early maturing players suggesting that they perceive greater physical and technical challenge when competing in maturity matched bio-banded match-play (18). Conversely, increasing the ratio of late to early maturing players may reduce the physical demand imposed on later maturing players during chronologically categorised (i.e., maturity mixed) SSG formats, given that late maturing players have stated they perceive less physical and technical load when contesting maturity matched bio-banded match-play (18). Therefore, consideration of player maturation status during the selection of players for SSG teams should be considered, and further research is required to establish the effect of low and high ratios of late and early maturing players on the physical, technical, tactical and psychological load of players during match-play.”

---

## [Decision Letter · Decision Letter 1]

19 Nov 2021

The effect of bio-banding on academy soccer player passing networks: Implications for relative pitch size

PONE-D-21-21543R1

Dear Dr. Towlson,

We’re pleased to inform you that your manuscript has been judged scientifically suitable for publication and will be formally accepted for publication once it meets all outstanding technical requirements.

Kind regards,

Emiliano Cè

Academic Editor

PLOS ONE

Additional Editor Comments (optional):

Reviewers' comments:

Reviewer's Responses to Questions

**Comments to the Author**

1. If the authors have adequately addressed your comments raised in a previous round of review and you feel that this manuscript is now acceptable for publication, you may indicate that here to bypass the “Comments to the Author” section, enter your conflict of interest statement in the “Confidential to Editor” section, and submit your "Accept" recommendation.

Reviewer #1: (No Response)

Reviewer #2: All comments have been addressed

2. Is the manuscript technically sound, and do the data support the conclusions?

Reviewer #1: Partly

Reviewer #2: Yes

3. Has the statistical analysis been performed appropriately and rigorously? 

Reviewer #1: Yes

Reviewer #2: Yes

4. Have the authors made all data underlying the findings in their manuscript fully available?

Reviewer #1: Yes

Reviewer #2: Yes

5. Is the manuscript presented in an intelligible fashion and written in standard English?

Reviewer #1: Yes

Reviewer #2: Yes

6. Review Comments to the Author

Reviewer #1: I would like to thanks the authors for the work provided to develop this article. I believe that the methodological issues are very interesting and with a great potential interest in practice. I want to consider the value of this methodological aspect into my review consideration. However, I still have some concern about practical application and how it is written. I believe that the practical application could be more practical and more innovative. The research findings can open to great innovative conclusions and to new ideas and approach to training. However the I believe that the practical applications ara a little it weak. Conversely, I have to acknowledge that this is my point of view and other can find these applications very innovative. I have to consider also the quality of data analysis and methods that can open to future investigations. For this reason, I believe that the paper can be suitable for publication in PLOS ONE. I believe that this article could be very interesting for his methodological aspect and can be of interest for researchers and practitioners. Even if I still have some dubts about the practical applications and conclusions provided, I have to acknowledge his methodological analysis. Therefore, I believe that this project could be consider for pubblication.

Reviewer #2: (No Response)

7. PLOS authors have the option to publish the peer review history of their article (what does this mean?). If published, this will include your full peer review and any attached files.

Reviewer #1: No

Reviewer #2: **Yes: **Jorge López-Fernández

---

## [Editor Report · Acceptance letter]

3 Dec 2021

PONE-D-21-21543R1 

The effect of bio-banding on academy soccer player passing networks: Implications of relative pitch size 

Dear Dr. Towlson:

I'm pleased to inform you that your manuscript has been deemed suitable for publication in PLOS ONE. Congratulations! Your manuscript is now with our production department. 

Kind regards, 

on behalf of

Professor Emiliano Cè 

Academic Editor

PLOS ONE